

# Hierarchy of degenerate stationary states in a boundary-driven dipole-conserving spin chain

Apoorv Srivastava[1⋆] and Shovan Dutta[2†]

**1** Department of Physics, Indian Institute of Space Science
and Technology, Thiruvananthapuram, Kerala 695547, India
**2** Raman Research Institute, Bangalore 560080, India

⋆ apoorvsri1909@gmail.com , † shovan.dutta@rri.res.in

## Abstract

Kinetically constrained spin chains serve as a prototype for structured ergodicity breaking in isolated quantum systems. We show that such a system exhibits a hierarchy of degenerate steady states when driven by incoherent pump and loss at the boundary. By tuning the relative pump and loss and how local the constraints are, one can stabilize mixed steady states, noiseless subsystems, and various decoherence-free subspaces, all of which preserve large amounts of information. We also find that a dipole-conserving bulk suppresses current in steady state. These exact results based on the flow in Hilbert space hold regardless of the specific Hamiltonian or drive mechanism. Our findings show that a competition of kinetic constraints and local drives can induce different forms of ergodicity breaking in open systems, which should be accessible in quantum simulators.



# 1 Introduction

In recent years, kinetic constraint has emerged as an important mechanism to induce layers of ergodicity breaking and subdiffusive transport in isolated many-body quantum systems [1,2]. In particular, such constraints can fragment conventional symmetry sectors into exponentially many disjoint blocks, which themselves can vary from ergodic to integrable in the same physical system [3–7]. Depending on how local the constraints are, one obtains different levels of fragmentation [8,9] and subdiffusion [10–15], whose signatures have been observed experimentally [16–20]. However, the physics is understood only for isolated, at most noisy [21], systems. Given that one can also engineer local drives in existing setups [22,23], it is particularly timely and intriguing to ask how such a motionally constrained system responds when driven by pump and loss at opposite ends. This is what we address in this paper.

We consider a paradigmatic model for fragmentation, namely a spin chain whose Hamiltonian conserves both a $U(1)$ charge and its dipole moment. These are usually taken as $\sum_i \hat{S}_i^z$ and $\sum_i i\hat{S}_i^z$, where $i$ labels consecutive sites. The dipole-moment conservation prohibits independent spin exchanges such as $\hat{S}_i^+ \hat{S}_j^-$, instead admitting pairwise terms such as $\hat{S}_{i-1}^+ \hat{S}_i^- \hat{S}_j^- \hat{S}_{j+1}^+$. One can equivalently think of a particle model where the particles hop in pairs to preserve their center of mass. Such correlated hops are realized on strongly tilted lattices [3,9,15–18,24–27] and in quantum Hall setups [28,29]. The allowed separation of the hops ($|i-j|$) controls the fragmentation. For normal spin chains without dipole symmetry, adding boundary drive has proved a fruitful way to explore transport properties and dissipative phase transitions [30]. In the presence of a bias, they typically reach a unique current-carrying steady state.

In contrast, we show that dipole conservation leads to a degenerate steady-state manifold, which varies qualitatively depending on the level of fragmentation and the nature of the drive. Furthermore, even though the drive breaks dipole conservation locally, the bulk constraints suppress current in steady state, instead giving rise to domain walls. When the Hamiltonian is not fragmented, a fraction of the symmetry sectors are immune to pump and loss at opposite ends and form decoherence-free subspaces (DFSs) [31,32] of total size $\sim \exp(\pi\sqrt{L/3})$, where $L$ is the number of spins. With pump and loss at both ends, these flow to one of $L-1$ mixed steady states due to a conserved quantum number or strong symmetry [33–35]. On the other hand, strong fragmentation stabilizes an exponentially large number of DFSs or noiseless subsystems (NSs) [32,36], where the bulk is shielded from the edges. All of these steady states preserve information [32,35] and represent breakdown of ergodicity in the full dynamics [37], which originates from the inability of the Hamiltonian to remove local excitations. Our exact results demonstrate that combining kinetic constraints with local dissipation is a promising route to stabilize different classes of degenerate manifolds in the same setup.

# 2 Model and realization

For simplicity we consider a spin-1/2 chain with nearest-neighbor exchanges, described by the Hamiltonian

$$\hat{H} = \sum_{i<j} \left( V_{i,j}\, \hat{S}_{i-1}^+ \hat{S}_i^- \hat{S}_j^- \hat{S}_{j+1}^+ + \text{h.c.} \right) + \hat{U}\left(\{\hat{S}_i^z\}\right), \tag{1}$$

where $\hat{U}$ is an arbitrary function of $\hat{S}_i^z$. We will use the spin and particle languages interchangeably and use 1 and 0 to represent spin-↑ (occupied site) and spin-↓ (empty site), respectively. We focus on two limits: (1) all separations $j-i$ are allowed and there is no fragmentation, i.e., $\hat{H}$ can couple any two Fock states with the same charge and dipole moment, and (2) $V_{i,j} = 0$ unless $j = i+1$, which is strongly fragmented and known as the pair-hopping model [3]. The dipole moment is conserved locally (in the same sense as used in Ref. [14]; see also [12,15])

in (2), which gives subdiffusion, but not in (1), which gives diffusion [14,15]. Our results are based on the connectivity in the Hilbert space and do not rely on the specific forms of $V$ or $\hat{U}$, although they will affect the timescales to reach steady state [9].

Such models describe the physics of interacting qubits on a strongly tilted lattice [3,9,15, 16,24,26] and interactions projected onto the lowest Landau level [28,29]. In particular, the fragmented limit is obtained as the leading-order term for tilted lattices with nearest-neighbor interactions [3,9,26] and for a Landau level on a thin torus [28]. For long-range interactions or away from the thin-torus limit, one finds other dipole-conserving quartic terms, including longer-range hops [15,16]. Nonetheless, here we allow only nearest-neighbor hops, as the DFS structure in the unfragmented case requires finite-range hops. This scenario may be simulated directly using four-qubit interactions among trapped ions [38,39] or digitally with four-qubit gates [40–43].

We assume the spin chain is subjected to incoherent pump and loss at its boundary, which can be modeled most simply by a Lindblad equation [44–46] for the density matrix $\hat{\rho}$,

$$\frac{\mathrm{d}\hat{\rho}}{\mathrm{d}t} = -\mathrm{i}\,[\hat{H},\hat{\rho}] + \sum_{k=1}^{4}\left(\hat{L}_k\hat{\rho}\hat{L}_k^\dagger - \frac{1}{2}\{\hat{L}_k^\dagger\hat{L}_k,\hat{\rho}\}\right), \tag{2}$$

with the jump operators $\hat{L}_1 = \hat{S}_1^+$, $\hat{L}_2 = \hat{S}_L^-$, $\hat{L}_3 = \sqrt{\gamma}\hat{S}_1^-$, and $\hat{L}_4 = \sqrt{\gamma}\hat{S}_L^+$ (we set $\hbar = 1$). The dimensionless rate $\gamma$ sets the pump-to-loss ratio. For $\gamma = 0$ one has pure pump at one end and pure loss at the other, which we will call "unipolar" drive for short.

Such a local Lindblad description is widely used for boundary-driven systems [30] and correspond to coupling the two end sites to infinite-temperature magnetization baths of opposite polarity. It can be physically justified for special types of Markovian baths [30,47–50]. However, most of our results are not contingent on this model, or even Markovianity, but only require some form of local injection and removal. Indeed, we use Eq. (2) only in Sec. 4 for quantifying dynamics in the presence of a strong symmetry.

While photon loss can be tuned in a lossy cavity [51], an incoherent pump was engineered by combining loss and two-photon drive in a superconducting circuit [22]. More generally, one may implement a pump by stochastically measuring $\hat{S}_1^z$ and applying a local $\pi$ pulse only if the outcome was ↓. In addition, the case of equal pump and loss ($\gamma = 1$) corresponds to pure dephasing and can be simulated with noisy $\sigma^x$ and $\sigma^y$ pulses as in Refs. [23,52].

## 3 Unfragmented with unipolar drive: Decoherence-free subspaces

Perhaps the most striking case is when the local nature of the drive and the hops gives rise to multiple DFSs even though the amplitudes $V_{i,j}$ are all nonzero and the model is unfragmented. With pump at the left end and loss at the right end, clearly an empty lattice ends up in $10\ldots0$ and a filled lattice in $1\ldots10$. In fact, there are $L-1$ frozen configurations with a single domain wall, of the form $1\ldots10\ldots0$, and it is possible to start from a Néel state such as $1010101$ and end up in $1111000$. Moreover, any superposition of these domain-wall states, which can be strongly entangled, is also frozen. These are not the only final states, however. Configurations of the form $1\ldots1010\ldots0$ do not evolve either, and pairs of states such as $110010$ and $101100$ oscillate between each other, forming a two-dimensional DFS [31]. More generally, there are symmetry sectors of $\hat{H}$, with a given particle number $N$ and dipole moment $D$, for which all states have the first site filled and the last site empty. Each of these sectors is unaffected by the pump/loss and becomes a separate DFS.

To identify these sectors, let us first define $D = \sum_i i n_i$ where $n_i$ are the site occupations. For a given $N$, one can increase $D$ in steps of 1 by starting from the domain-wall state $1\ldots10\ldots0$,

moving the $N$-th particle to the right until it reaches the right end, then moving the $(N-1)$-th particle to the right, and so on. The last site is empty only during the first sweep, when the state has the form

$$\underbrace{1\ldots1}_{N-1}\,\underbrace{0\ldots0}_{p}\,1\,\underbrace{0\ldots0}_{\geq1}, \tag{3}$$

where $p$ is a nonnegative integer. Starting from such a state, the first site can become 0 under $\hat{H}$ if the first $N-1$ particles can shift to the right, which requires the last particle to hop $N-1$ sites to the left. Due to the hard-core constraint, this is possible only for $p \geq N$. Hence, the decoherence-free sectors have $p \leq N-1$, corresponding to the "root" configurations

$$\underbrace{1\ldots1}_{\geq0}\,1\,\underbrace{1\ldots1}_{p}\,\underbrace{0\ldots0}_{p}\,1\,\underbrace{0\ldots0}_{\geq1}, \tag{4}$$

which reduce to a single domain wall for $p=0$. For $p \geq 2$, only the red-colored sites evolve under the Hamiltonian, forming an active center surrounded by frozen wings. The number of DFSs is given by the number of distinct choices for $(N,p)$, which amounts to $N_{\text{DFS}} = \lfloor L^2/4 \rfloor$, whereas the number of $(N,D)$ sectors is given by $N_{\text{sector}} = (L^3 + 5L + 6)/6$ (see Appendix A). Thus, only $O(1/L)$ of all the symmetry sectors become DFSs. The size of a DFS is set by the number of arrangements within the active block of the root configuration that have the same dipole moment. This is equivalent to finding integer solutions to the equation

$$x_1 + x_2 + \cdots + x_p = p(p-1)/2 + 2p, \tag{5}$$

where $1 \leq x_1 < x_2 < \cdots < x_p \leq 2p$. Defining $y_i := x_i - i$ the equation becomes

$$y_1 + y_2 + \cdots + y_p = p, \quad \text{with} \quad 0 \leq y_1 \leq y_2 \leq \cdots \leq y_p \leq p, \tag{6}$$

which are all integer partitions of $p$. Hence, the DFS size grows as $d_p \sim \exp(\pi\sqrt{2p/3})/(4\sqrt{3}p)$ for $p \gg 1$ [53]. The total number of all decoherence-free states scales as (see Appendix A)

$$d_{\text{DFS}} = \sum_{p,N} d_p \sim \frac{\sqrt{3}}{\pi^2}\exp\left(\pi\sqrt{L/3}\right)\left[1 + O(1/\sqrt{L})\right]. \tag{7}$$

Not all initial states can reach a given DFS. This is because the dynamics still preserve a quantum number, even though the conservation of $N$ and $D$ are broken. If we call the leftmost site $i=0$, the pump does not alter $D$ and the loss at the right end decreases $D$ only in steps of $L-1$. Thus, they conserve the integer $D_{\text{mod}} := D \pmod{L-1}$, splitting the dynamics into $L-1$ blocks of roughly equal size. The resulting flow in Hilbert space is shown for $L = 10$ in Fig. 1, where each node represents a symmetry sector of the Hamiltonian, labeled by $(N,D)$, and each arrow represents a pump or loss event connecting two sectors. All paths flow to one of the DFSs, which are colored red. Conversely, the blue-colored "sources" have the first site empty and the last site filled—these would become DFSs if one were to swap the pump and loss. As exchanging left and right changes $D$ to $(L-1)N-D$, the sources for a given $D_{\text{mod}}$ turn into the sinks for $L-1-D_{\text{mod}}$. For $D_{\text{mod}} = 0$ the two are balanced.

We make two observations: First, the arrows show classical paths as they denote either a pump or a loss event. Therefore, starting with a definite $(N,D)$ one always ends up in a unique DFS. In order to reach a superposition of multiple DFSs, such as two of the domain-wall states, one must necessarily start from a superposition of multiple nodes. So the dissipative drive does not provide a straightforward way to produce structured entanglement. Of course, a product state can still end up in a large DFS with volume-law entanglement [2–5,8,11,54–57]. Second, in all of the DFSs the first and last sites are frozen at 1 and 0, respectively. Thus, there is no net current in steady state and any unidirectional flow must be transient.

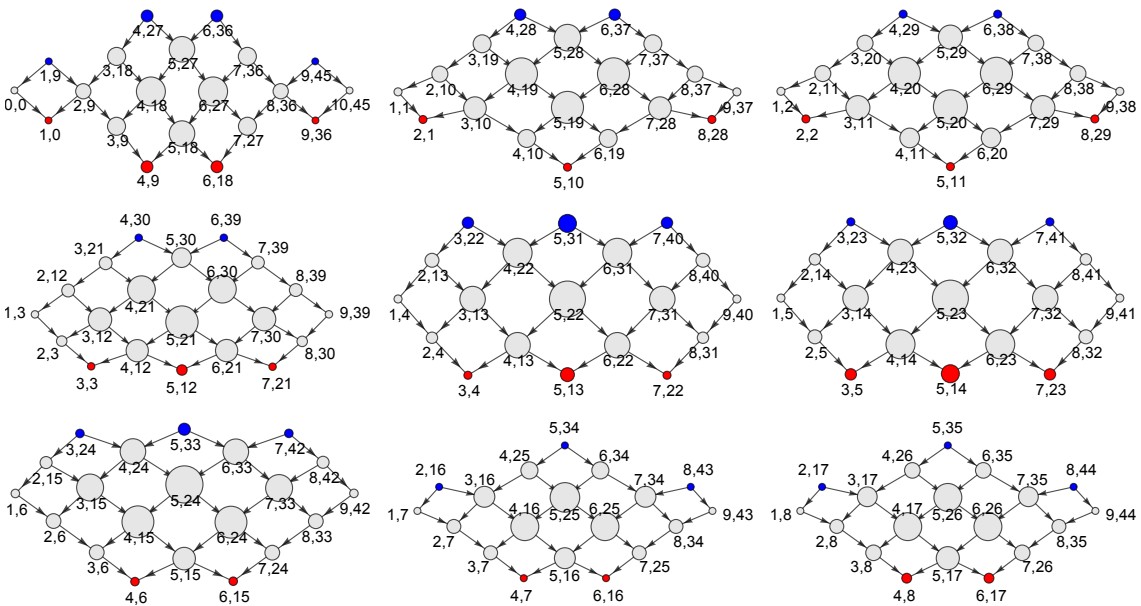

Figure 1: Flow in the Hilbert space of a dipole-conserving spin-1/2 chain with $L = 10$ sites, driven by incoherent pump at the first site, $i = 0$, and loss at the last site, $i = 9$. Each node represents a symmetry sector of the Hamiltonian labeled by the charge $N = \sum_i n_i$ and dipole moment $D = \sum_i i n_i$, where $n_i \in \{0, 1\}$ are the site occupations. The area of a node is proportional to its dimension, and the arrows represent either a pump or a loss event. The flow is split into $L - 1$ blocks characterized by $D_{\mathrm{mod}} := D$ (mod $L - 1$), which varies from 0 to 8 from the top left to the bottom right. Inside a block, the flow starts from one of the "sources" (blue nodes) at the top and ends in one of the decoherence-free subspaces (red nodes) at the bottom.

## 4 Unfragmented with bipolar drive: Strong symmetry

With pump and loss at both ends, the arrows in Fig. 1 become bidirectional. Hence, the DFSs are no longer stable. However, the conservation of $D_{\mathrm{mod}}$ still holds. Such a quantum number, which is conserved by both the Hamiltonian and the dissipation, is called a strong symmetry in the context of Lindblad dynamics [33–35]. In the absence of further symmetries, each block of $D_{\mathrm{mod}}$ reaches a unique, mixed steady state. The structure of the steady state also follows from the flow in Fock space. In particular, we see that the Fock states can be arranged vertically in layers such that the pump and loss couple only adjacent layers. If, in addition, the pump rate at the first site equals the loss rate at the last site (and vice versa) [58–60], all transitions from a given layer to the one below it have the same rate (set to 1), and all upward transitions have rate $\gamma$ [see Eq. (2)]. Such a flow diagram implies a steady state with detailed balance [61], where all configurations in a given layer are equally likely and the weights in successive layers differ by a factor of $\gamma$, as sketched in Fig. 2(a) for $L = 6$ and $\gamma = 0.5$. Thus, each $(N, D)$ sector heats to infinite temperature, and their relative weights ensure no net flow between any two Fock states. To further characterize these weights, note that the successive nodes in a given layer differ in $(N, D)$ by $(2, L - 1)$. Thus, all of them share the same integer

$$Q := N - 2(D - D_{\mathrm{mod}})/(L - 1), \tag{8}$$

which increases in steps of 1 from the top layer to the bottom layer. The steady state is given by the Gibbs ensemble $\hat{\rho} \propto \exp(\mu \hat{Q})$ with $\mu = \ln(1/\gamma)$. This holds irrespective of the form of the Hamiltonian as long as it couples all Fock states with the same $N$ and $D$.

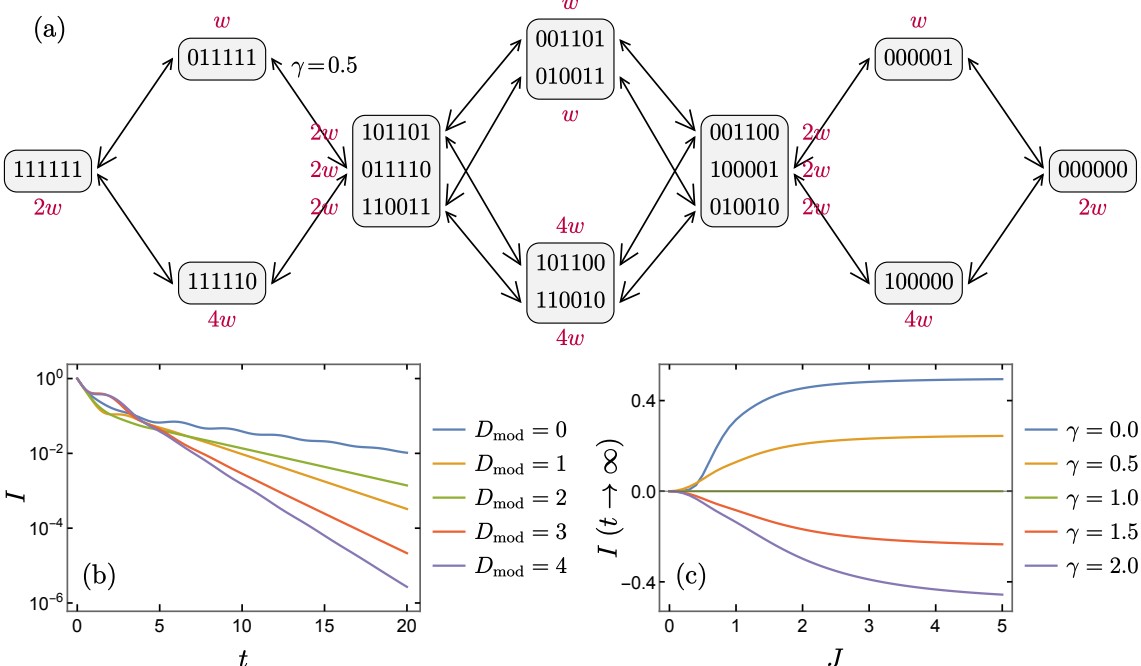

Figure 2: (a) Flow in the $D_{\mathrm{mod}} = 0$ block for $L = 6$ in the presence of pump and loss at both ends. Downward transitions are caused by injection at the left end or loss at the right end (with rate 1), whereas upward transitions are caused by loss at the left end or injection at the right end with rate $\gamma = 0.5$. The red labels show steady-state population of each configuration (up to normalization), satisfying detailed balance. (b) Overall rate at which particles are injected into the first site as a function of time for the Lindblad dynamics in Eq. (2) with $L = 6$, $\gamma = 0.5$, $V_{i,j} = 1$, and $\hat{U} = 0$, starting from a single particle at site $i = D_{\mathrm{mod}}$. (c) The same rate in steady state after adding the symmetry-breaking term $\delta\hat{H} = -J \sum_i \hat{S}_i^+ \hat{S}_{i+1}^- + \text{h.c.}$ to the Hamiltonian.

As one might expect, there is no net current in the system in such an infinite-temperature steady state. This is confirmed explicitly for the Lindblad dynamics [Eq. (2)] for small systems. As shown in Fig. 2(b) for $V_{i,j} = 1$, the overall rate of injection from outside into the first site, $I = 1 - \langle \hat{n}_0 \rangle - \gamma \langle \hat{n}_0 \rangle$, decays exponentially with time for all values of $D_{\mathrm{mod}}$. The decay rate is set by the Liouvillian gap which would depend on $V_{i,j}$, $\gamma$, and $L$ [62,63]. We also find that the current is restored by breaking the dipole symmetry of the Hamiltonian, e.g., with a magnetic field in the $x$ direction or by independent nearest-neighbor hops. Figure 2(c) shows how the steady-state current grows monotonically with the strength of the latter.

## 5  Fragmented with unipolar drive: Decoherence-free subspaces

Next we turn to the pair-hopping model [3], for which $V_{i,j} = 0$ unless $j = i+1$ and each $(N, D)$ sector is strongly fragmented. Due to the nearest-neighbor constraint, any state composed of sequences of three or more 1's or 0's does not evolve under the Hamiltonian. Thus, with pump at left end and loss at right end, we obtain exponentially many frozen states of the form

$$\underbrace{1\ldots1}_{\geq 3}\,\underbrace{0\ldots0}_{\geq 3}\,\underbrace{1\ldots1}_{\geq 3}\ldots\underbrace{0\ldots0}_{\geq 3}\,. \tag{9}$$

This is in stark contrast with the unfragmented case, where only $2L - 4$ configurations, with one or three domain walls, were frozen. Furthermore, as we show below, there is a much

larger set of decoherence-free states, including $2^{L-4}$ configurations of the form

$$11 \boxed{\phantom{xxxxxxxxxxxxxx}} 00 \tag{10}$$

where the box can contain an arbitrary binary sequence. These states are grouped into exponentially many DFSs given by the fragmentation structure of the Hamiltonian [3].

The form in Eq. (10) is not necessary for a state to be decoherence free. All we need is that the first site is frozen at 1 and the last site is frozen at 0. Thus, it suffices to discard only those configurations of the form $1 \boxed{\phantom{xxx}} 0$ for which this is not true. To see how this works, note that the dynamics under the Hamiltonian consist of changing four consecutive sites from 1001 to 0110 and vice versa. Hence, the first spin can flip only if the first four sites are 1001, which means we should omit states of the form $1001 \boxed{\phantom{xxx}} 0$. However, these can in turn arise from $101001 \boxed{\phantom{xxx}} 0$ and $1000110 \boxed{\phantom{xxx}} 0$, and so on. This hierarchy of decohering states is generated by recursively applying the expansion rules

$$
\begin{array}{cc}
\underline{01} & \underline{10} \\
\swarrow \quad \searchdownarrow & \swarrow \quad \searrow \\
10\underline{01} \quad 00\underline{110} & 01\underline{10} \quad 1100\underline{01}
\end{array}
\tag{11}
$$

to the last two digits (underlined) of the lead sequence. The resulting "1001 family" of states have the form

$$\Uparrow \dots \Uparrow 0 \Downarrow \dots \Downarrow 1 \Uparrow \dots \Uparrow 0 \Downarrow \dots \Downarrow 1 \cdots \cdots 1 \Uparrow \dots {\color{red}\Uparrow\Downarrow} \boxed{\phantom{xxxx}} 0 \tag{12a}$$

$$\text{or} \quad \Uparrow \dots \Uparrow 0 \Downarrow \dots \Downarrow 1 \Uparrow \dots \Uparrow 0 \Downarrow \dots \Downarrow 1 \cdots \cdots 0 \Downarrow \dots {\color{red}\Downarrow\Uparrow} \boxed{\phantom{xxxx}} 0, \tag{12b}$$

where the collective spins $\Uparrow$ and $\Downarrow$ stand for 10 and 01, respectively [3]. The last two collective spins (colored red) represent the active edge of the sequence—no site to their left can change without first swapping them. Such a swap produces an ancestor in the family tree. As we detail in Appendix B, the total number of states in this family, for which the first site can eventually flip, is $N_{1001} \approx 2^{L-2}/5$. Similarly, the last site can flip if the last four sites become 0110, which in turn gives rise to a family of states expanding to the left of the same size as $N_{1001}$. The set of all decoherence-free states is obtained by discarding members of either of the two families, which leaves a manifold of total size (see Appendix B)

$$d_{\text{DFS}} \approx (2/5)^2 \times 2^L. \tag{13}$$

In summary, the bulk of the system remains strongly fragmented and a finite fraction of the Hilbert space becomes decoherence free, as opposed to a vanishingly small fraction in the unfragmented case [Eq. (7)]. As in the latter, there is no net current in steady state.

## 6 Fragmented with bipolar drive: Noiseless subsystems

With pump and loss at both ends, no state is unaffected by the drive, so we do not have a DFS. However, we can insert a "blockade" sequence [3, 8, 9, 57] of 1111 or 0000 at either end to shield the bulk from the boundary. These are initial states of the form

$$\boxed{0/1}\, 111 \boxed{1 \phantom{xxxxxxx} 0}\, 000 \boxed{0/1}. \tag{14}$$

From our discussion in Sec. 5 and the nearest-neighbor constraint, it follows that the first four and last four sites cannot change under $\hat{H}$. Thus, information in the bulk is preserved, which

constitutes a noiseless subsystem (NS) [32,36], while the end sites continue to flip between 0 and 1, taking the system to a mixed steady state [see Eq. (2)]

$$\hat{\rho} \propto \left(\gamma|0\rangle\langle 0| + |1\rangle\langle 1|\right) \otimes \left|\psi_j^{\text{bulk}}\right\rangle\left\langle\psi_j^{\text{bulk}}\right| \otimes \left(|0\rangle\langle 0| + \gamma|1\rangle\langle 1|\right), \tag{15}$$

where $|\psi_j^{\text{bulk}}\rangle$ is an eigenstate of the bulk Hamiltonian. In fact, the number of bulk states that are immune to the end flips is much larger than the $2^{L-6}$ configurations of the type in Eq. (14). All we require is that the two end sites are not altered by $\hat{H}$. As we saw in Sec. 5, the first site is not altered from 1 for all states outside the 1001 family [Eq. (12)], and the last site is not altered from 0 for all states outside the 0110 family [Eq. (B.3)]. One has to also exclude states for which the Hamiltonian can flip the first site from 0 to 1 or the last site from 1 to 0. These are obtained by swapping 1's with 0's in the two previous families. By symmetry the four groups are of equal size. They comprise a total of approximately $(16/25)\,2^{L-2}$ configurations, as sketched below (with arrows indicating the direction in which the family expands).

$$
\begin{gathered}
\text{(Venn diagram: } A: 1001\rightarrow,\ B: \leftarrow 0110,\ D: \leftarrow 1001,\ C: 0110\rightarrow\text{)}
\end{gathered}
\qquad
\begin{aligned}
d_A &= d_B = d_C = d_D \approx 2^{L-2}/5\,, \\
d_{A\cap B} &= d_{C\cap D} \approx d_{A\cap D} = d_{B\cap C} \approx 2^{L-2}/25\,.
\end{aligned}
\tag{16}
$$

Hence, the dimension of the NS is given by $d_{\text{NS}} \approx (9/25)\,2^{L-2}$. These bulk states follow the fragmentation of the Hamiltonian. As before, there is no net current in steady state.

# 7 Conclusion

We studied how a dipole-conserving spin-1/2 chain responds when driven by incoherent pump and loss at opposite ends. We found that, even though the drive locally breaks both charge and dipole conservation, the kinetic constraints in the bulk suppress current at late times, driving the system to a degenerate steady-state manifold. By analyzing the flow in Hilbert space, we were able to derive exact results that fully characterize the manifold and does not depend on the precise form of the Hamiltonian or the details of how the pump and loss are implemented. We showed the nature of the manifold can be tuned across a wide range of possibilities (DFS, NS, mixed steady states) by varying whether the dipole conservation is local or global and whether the pump and loss act on opposite ends or both ends. Such flexibility is highly unusual for Markovian systems which generically evolve to a unique steady state [35,37]. Our main findings should be accessible in present-day quantum simulators (see Sec. 2). Our results show that combining kinetic constraints with local dissipative drives is a promising avenue to induce different types of ergodicity breaking in open quantum dynamics.

We conclude with three remarks for future work. First, while we have focused on steady states, past studies have established an intimate link between multipole conservation and sub-diffusion in isolated systems [10–15], which begs the question of how the subdiffusive transport controls the approach to steady state [30]. Second, although we have taken a boundary-driven spin-1/2 chain for simplicity, we expect our conclusions to generalize to other cases. Nonetheless, it would be particularly interesting to see whether one can stabilize a current-carrying steady state by increasing the local dimension [8,9,15,17,27,57] or by driving interior sites [64]. Third, as dipole-conserving models are fragmented in a product basis [7–9], most of our findings carry over to a dipole-conserving classical exclusion process [10,42,43,65–70].

But, for the same reason, it is difficult to create entangled states by a local drive (see Sec. 3). It would be valuable to explore if one can circumvent this drawback in systems that are fragmented in an entangled basis [7, 21] or by using fermionic loss [71].

## Acknowledgments

We thank Sanjay Moudgalya for useful discussions.

**Funding information** This work was supported by intramural funds of the Raman Research Institute.

## A  Counting decoherence-free states in the unfragmented case

For a given particle number $N$, the dipole moment $D$ is minimum for the state $1\ldots10\ldots0$ and increases in steps of 1 as we move each particle $L-N$ sites to the right, one at a time. Hence, the number of distinct $(N, D)$ sectors is given by $N_{\text{sector}} = \sum_{N=0}^{L} 1 + N(L-N) = (L^3 + 5L + 6)/6$.

The number of disjoint DFSs can be found by counting configurations of the type in Eq. (4) with $p \leq N-1$ that can fit in $L$ sites, i.e., $N + p + 1 \leq L$. Thus,

$$N_{\text{DFS}} = \sum_{N=1}^{L-1} \sum_{p=0}^{\min(N,L-N)-1} 1 = \left\lfloor \frac{L^2}{4} \right\rfloor. \tag{A.1}$$

As shown in Eqs. (5) and (6), the size of a DFS is given by the number of integer partitions of $p$, denoted by $d_p$. Hence, the total count of all decoherence-free states amounts to

$$d_{\text{DFS}} = \sum_{N=1}^{L-1} \sum_{p=0}^{\min(N,L-N)-1} d_p = \sum_{p=0}^{\lfloor L/2 \rfloor - 1} \sum_{N=p+1}^{L-p-1} d_p = \sum_{p=0}^{\lfloor L/2 \rfloor - 1} (L - 2p - 1) d_p. \tag{A.2}$$

Using $d_p \sim \exp(\pi\sqrt{2p/3})/(4\sqrt{3}p)$ for $p \gg 1$ [53], we find

$$d_{\text{DFS}} \sim \int_1^{L/2} dp \, (L - 2p) \frac{e^{\pi\sqrt{2p/3}}}{4\sqrt{3}p} = F(L) - F(2), \tag{A.3}$$

where

$$F(x) := \frac{L}{2\sqrt{3}} \, \text{Ei}\left(\pi\sqrt{\frac{x}{3}}\right) - e^{\pi\sqrt{x/3}}\left(\frac{\sqrt{x}}{2\pi} - \frac{\sqrt{3}}{2\pi^2}\right), \tag{A.4}$$

and Ei is the exponential integral function [72]. Using the asymptotic expansion

$$\text{Ei}(x) \sim \frac{e^x}{x}\left(1 + \frac{1!}{x} + \frac{2!}{x^2} + \cdots\right), \tag{A.5}$$

gives the scaling

$$d_{\text{DFS}} \sim \frac{\sqrt{3}}{\pi^2} e^{\pi\sqrt{L/3}}\left[1 + \frac{\sqrt{3}}{\pi\sqrt{L}} + O\left(\frac{1}{L}\right)\right]. \tag{A.6}$$

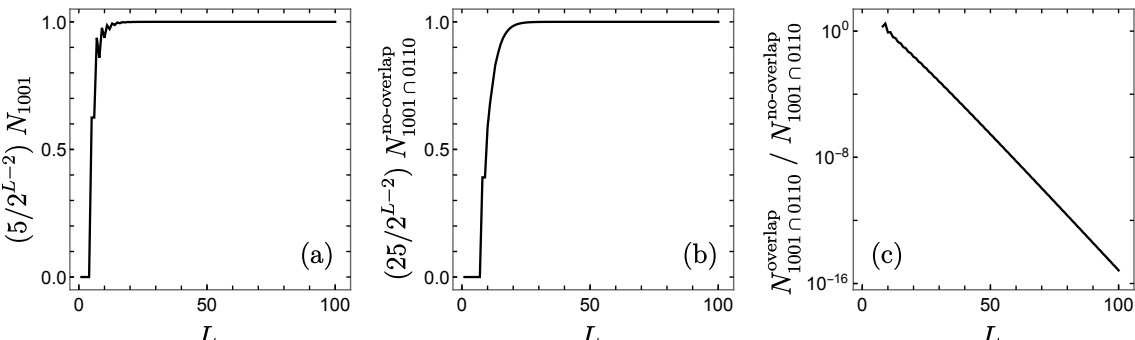

Figure 3: (a) Number of states in the 1001 family, having configurations of the form in Eq. (12) for which the first site can flip from 1 to 0 through a sequence of swaps between 1001 to 0110. (b) Number of states of the form in Eq. (B.4), for which both the first site and the last site can flip from 1 and 0, respectively. (c) Number of such unstable states for which the lead sequences on either side overlap, in comparison.

## B  Counting decoherence-free states in the fragmented case

First we count the 1001 family given by the configurations in Eq. (12). The lead sequence of such a state can be labeled the number of collective spins, $n \geq 2$, and the number of domain walls of the form $\Uparrow 0 \Downarrow$ and $\Downarrow 1 \Uparrow$, $k \leq n-2$. For a finite system they also satisfy $2n+k \leq L-1$. There are $\binom{n-2}{k}$ ways of placing the domain walls and $2^{L-1-2n-k}$ arrangements of the sites in the box for a given lead sequence [see Eq. (12)]. Hence, the total number of states is given by

$$N_{1001} = \sum_{n=2}^{\lfloor (L-1)/2 \rfloor} \sum_{k=0}^{\min(n-2,L-1-2n)} \binom{n-2}{k} 2^{L-1-2n-k} \approx \frac{2^{L-2}}{5}. \tag{B.1}$$

For odd values of $L \geq 7$, one has to add states for which the lead sequence spans the lattice, ending with 0110. These have an odd number of domain walls that can placed in $(L-k)/2-2$ positions, yielding a total count of approximately $0.13 \times 1.32^L$.

Next we turn to the 0110 family, for which the last site can eventually flip from 0 to 1. Like the 1001 family, the states are generated by recursively applying the expansion rules

$$
\begin{array}{cc}
\underline{01} & \underline{10} \\
\swarrow \quad \searrow & \swarrow \quad \searrow \\
\underline{01}10 \quad \underline{10}011 & \underline{10}01 \quad \underline{01}100
\end{array}
\tag{B.2}
$$

to the leftmost two digits (underlined), which leads to the form

$$1\ \boxed{\phantom{xxxx}}\ \Uparrow\Downarrow \ldots \Downarrow 1 \cdots \cdots 0 \Downarrow \ldots \Downarrow 1 \Uparrow \ldots \Uparrow 0 \Downarrow \ldots \Downarrow 1 \Uparrow \ldots \Uparrow \tag{B.3a}$$

$$\text{or}\quad 1\ \boxed{\phantom{xxxx}}\ \Downarrow\Uparrow \ldots \Uparrow 0 \cdots \cdots 0 \Downarrow \ldots \Downarrow 1 \Uparrow \ldots \Uparrow 0 \Downarrow \ldots \Downarrow 1 \Uparrow \ldots \Uparrow. \tag{B.3b}$$

By symmetry the two families have equal size, $N_{0110} = N_{1001} \approx 2^{L-2}/5$ [see Fig. 3(a)].

The number of states shared between the two is dominated by configurations of the form

$$\Uparrow \ldots \Uparrow 0 \Downarrow \ldots \Downarrow 1 \cdots \cdots \Uparrow\Downarrow\ \boxed{\phantom{xxxxxx}}\ \Downarrow\Uparrow \cdots \cdots 0 \Downarrow \ldots \Downarrow 1 \Uparrow \ldots \Uparrow, \tag{B.4}$$

where the lead sequences do not overlap. Labeling the number of collective spins in these two sequences by $m$ and $n$, and the number of domain walls by $p$ and $q$, we find [as in Eq. (B.1)]

$$N_{1001 \cap 0110}^{\text{no-overlap}} = \sum_{\substack{m,n \geq 2 \\ 2(m+n)+p+q \leq L}} \sum_{p=0}^{m-2} \sum_{q=0}^{n-2} \binom{m-2}{p}\binom{n-2}{q} 2^{L-2(m+n)-p-q} \approx \frac{2^{L-2}}{25}. \tag{B.5}$$

For the other shared states, the lead sequences overlap by at most 4 sites. This is because the patterns 1001 and 0110 only appear at the active edges (red collective spins). As we show in Fig. 3(c), this overlapping set is exponentially small compared to the non-overlapping set. Hence, the total size of the two families can be estimated as

$$N_{1001\cup0110} \approx N_{1001} + N_{0110} - N_{1001\cap0110}^{\text{no-overlap}} \approx (9/25)\, 2^{L-2}\,. \tag{B.6}$$

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
