# Peer review of "Hierarchy of degenerate stationary states in a boundary-driven dipole-conserving spin chain"

_SciPost Physics, doi:SciPost Phys. 18, 111 (2025)_

## Round 1 · Referee Report · Anonymous (Referee 1) · 2024-12-19

Report

The authors investigate how kinetic constraints in boundary driven dipole-conserving spin chains can produce a hierarchy of degenerate stationary states that break ergodicity, in contrast to conventional boundary driven spin chains which generically reach a unique steady state. By analysing both fragmented and unfragmented cases under different driving regimes (unipolar and bipolar), they demonstrate how decoherence-free subspaces can be stabilized even in the presence of symmetry-breaking drives. The results are exact, based on the symmetry and structure of the Hamiltonian rather than specific forms of dynamics. The paper is clear and well written, timely adding valuable insights into ergodicity-breaking phenomena in open quantum systems, relevant to fields such as quantum thermodynamics and quantum error correction. With some minor corrections which I will elaborate on below, I believe the work should be published.

1. It would be helpful for the reader if the values of N_DFS, N_sector and d_DFS were derived in an appendix. I appreciate they may be simple derivations, but I still think it's worth including.

2. What's the intuition behind the root configurations given in section 3? Once given, they can be understood but I'm curious as to how the authors arrived at them.

3. Unless I’ve misunderstood, the statement that sequences of 3 or more consecutive 1's or 0's do not evolve under the Hamiltonian is incorrect. For example, the state 111001 can evolve to 110110, 1000110 can evolve to 1001001 etc. I understand that the sequence in isolation can’t evolve, but this isn’t quite what is stated.

4. The statement that any state beginning with 111 and ending in 000 is unaffected by the drives is true and intuitive but not immediately obvious. The reasoning given for which states to discard (Eq's 10 and 11) could be used to explain this but is given after the statement, so needs some slight restructuring. The same goes for the discussion about shielding the bulk with 111 and 000.

5. “By varying whether the dipole conservation is local or global", this statement is incorrect. In both cases of the allowed separation for the hops, the dipole moment is conserved. The allowed hop separation is what changes from local to global, which controls the fragmentation.

6. In the first row of Figure 1, arrows to the right denote losses and arrows to the left denote pumping. Why is this switched for rows 2 and 3?

Recommendation

Ask for minor revision

  • validity: -
  • significance: -
  • originality: -
  • clarity: -
  • formatting: -
  • grammar: -

Author:  Shovan Dutta  on 2025-02-20  [id 5234]

(in reply to Report 1 on 2024-12-19)

We have incorporated the suggestions of the referee into the revised version of the manuscript:

>> 1. It would be helpful for the reader if the values of N_DFS, N_sector and d_DFS were derived in an appendix. I appreciate they may be simple derivations, but I still think it's worth including.

We have included a derivation of these quantities in a separate appendix.

>> 2. What's the intuition behind the root configurations given in section 3? Once given, they can be understood but I'm curious as to how the authors arrived at them.

While we had initially arrived at these configurations through a process of trial and error, this question of the referee prompted us to think of a more logical approach. We have added a step-by-step reasoning to deduce the configurations by considering a mechanism that yields all possible dipole-moment sectors for a given particle number.

>> 3. Unless I’ve misunderstood, the statement that sequences of 3 or more consecutive 1's or 0's do not evolve under the Hamiltonian is incorrect. For example, the state 111001 can evolve to 110110, 1000110 can evolve to 1001001 etc. I understand that the sequence in isolation can’t evolve, but this isn’t quite what is stated.

We have revised this sentence to "any state composed of sequences of three or more 1's or 0's does not evolve under the Hamiltonian."

>> 4. The statement that any state beginning with 111 and ending in 000 is unaffected by the drives is true and intuitive but not immediately obvious. The reasoning given for which states to discard (Eq's 10 and 11) could be used to explain this but is given after the statement, so needs some slight restructuring. The same goes for the discussion about shielding the bulk with 111 and 000.

We agree with the referee that these statements were unclear. We have reworded the sentences such that they are clearly explained or follow from previous discussion.

>> 5. “By varying whether the dipole conservation is local or global", this statement is incorrect. In both cases of the allowed separation for the hops, the dipole moment is conserved. The allowed hop separation is what changes from local to global, which controls the fragmentation.

We have added a sentence in Sec. 2 explaining what we mean by the dipole moment being conserved locally when the hops are adjacent, as we discussed in our previous reply.

>> 6. In the first row of Figure 1, arrows to the right denote losses and arrows to the left denote pumping. Why is this switched for rows 2 and 3?

We thank the referee for catching this oversight. We were using Mathematica’s default function for drawing graphs which caused the switching. We have updated the figure such that the pump-loss directions are the same for all graphs.

Author:  Shovan Dutta  on 2024-12-20  [id 5056]

(in reply to Report 1 on 2024-12-19)
Category:
answer to question
correction
pointer to related literature

We sincerely thank the Referee for the positive and timely report. We will address all of the points in our full response. For now, we would like to clarify that: (i) For the 3rd point, the statement is indeed not correctly stated in the manuscript. It should say that "any state composed solely of sequences of three or more 1's or 0's does not evolve under the Hamiltonian." (ii) For the 5th point, the Referee is correct that the dipole moment D is conserved in both cases. However, when the hops are adjacent, D is also conserved locally (in the same sense as used in Ref. [14]; see also Refs. [12, 15]), which gives subdiffusion, whereas for large separation, D is only conserved globally, which gives diffusion [14, 15]. We will improve this wording in the manuscript.

---

## Round 1 · Referee Report · Anonymous (Referee 2) · 2025-1-17

Report

The authors of this paper studied stationary states of a rich, kinetically constrained spin chain setup that conserves both the charge and the dipole moment. Such a setup has been of significant interest in recent times. In the absence of a dipole-conserving term, spin chain setup has been studied from the boundary-driving context, where one usually expects a unique steady state. In this paper, the authors tried to analyze the impact of the dipole-conserving term and interestingly found degenerate steady-state manifold and suppression of current. This is even after boundary driving at the two ends, which locally breaks the dipole conservation. The authors further showed that the nature of such degenerate manifolds can be manipulated by making the dipole conserving term either local or global, and the nature of the boundary driving at the two ends. The authors provide a convincing Hilbert space flow framework to establish their results and further pinpoint how not all initial states can give rise to degenerate steady states. Such a model therefore gives rise to a rich structure to the non-equilibrium steady states even in the Markovian/Lindblad framework, which is typically unusual. Overall, the paper contains a rich set of results. The manuscript is also written very clearly, and the presentation is nice. I therefore recommend this paper for publication.

Recommendation

Publish (easily meets expectations and criteria for this Journal; among top 50%)

  • validity: high
  • significance: good
  • originality: good
  • clarity: good
  • formatting: perfect
  • grammar: excellent

Author:  Shovan Dutta  on 2025-02-20  [id 5235]

(in reply to Report 2 on 2025-01-17)

We thank the referee for the concise summary and the positive recommendation.

---

## Round 2 · Referee Report · Anonymous (Referee 1) · 2025-3-6

Report

I appreciate the comprehensive response provided by the authors addressing all of my queries and concerns. Now the appropriated revisions have been implemented, I recommend this paper for publication. However, there is one small formatting error in which figure 3 appears within the references rather than in Appendix B. This can be trivially corrected so does not need another round of review to address.

Recommendation

Publish (easily meets expectations and criteria for this Journal; among top 50%)

  • validity: -
  • significance: -
  • originality: -
  • clarity: -
  • formatting: -
  • grammar: -

Author:  Shovan Dutta  on 2025-03-07  [id 5270]

(in reply to Report 1 on 2025-03-06)

We thank the referee for the positive recommendation. We are unable to spot the formatting error — in the revised version on arXiv, Figure 3 appears within Appendix B on page 10. We apologize for any oversight on our part.

Anonymous on 2025-03-07  [id 5271]

(in reply to Shovan Dutta on 2025-03-07 [id 5270])
Category:
answer to question

Upon checking, the formatting error I found is in the link https://shovanduttaorg.wordpress.com/wp-content/uploads/2025/02/2411.03309_diff_v1v2.pdf rather than the arXiv version so this isn't an issue, apologies for my oversight.

---

## Round 2 · Author Response

Dear Editor,

We thank the referees for their careful review, useful suggestions, and positive recommendations. We have addressed all of the points raised by the referees in our revised manuscript. We hope it is now suitable for publication in SciPost Physics.

Please find below a list of the changes made. We provide a point-by-point response in our reply to each referee report. We also provide a version of the manuscript with the changes highlighted at: https://shovanduttaorg.wordpress.com/wp-content/uploads/2025/02/2411.03309_diff_v1v2.pdf.

Yours sincerely,
Apoorv Srivastava and Shovan Dutta.

---

## Round 2 · List of Changes

*Added a new appendix (Appendix A) to include derivations of state counting.
*Added a sentence in the first paragraph of Sec. 2 to explain local vs global conservation of the dipole moment.
*Expanded the first paragraph of Sec. 3 to include a step-by-step reasoning to arrive at the root configurations.
*Updated Figure 1 to make the pump and loss directions consistent across all the graphs.
*Corrected the second line of Sec. 5 about frozen states.
*Reworded the sentence preceding Eq. (10) on decoherence-free states.
*Updated Eq. (14) and the surrounding text on blockades such that they follow from past discussion.
*Updated published preprints.
*Added new references that are pertinent to the study:
-Ref. [57]: arXiv:2408.10321
-Ref. [69]: Physical Review E 110, 024119 (2024)
-Ref. [70]: Journal of Statistical Mechanics, 023201 (2025)
-Ref. [72]: NIST Handbook of Mathematical Functions, Cambridge University Press, New York (2010).

---

## Editorial Decision

published